# Inhibition of Metalloproteinases Extends Longevity and Function of In Vitro Human iPSC-Derived Skeletal Muscle

**DOI:** 10.3390/biomedicines12040856

**Published:** 2024-04-12

**Authors:** Natali Barakat, Himanshi Jangir, Leandro Gallo, Marcella Grillo, Xiufang Guo, James Hickman

**Affiliations:** 1NanoScience Technology Center, University of Central Florida, 12424 Research Parkway, Suite 400, Orlando, FL 32826, USA; na367569@ucf.edu (N.B.); himanshi.jangir@ucf.edu (H.J.); leandro.gallo@ucf.edu (L.G.); mgrillo@umich.edu (M.G.); xguo@mail.ucf.edu (X.G.); 2Department of Chemistry, University of Central Florida, Orlando, FL 32828, USA

**Keywords:** metalloproteinases, extracellular matrix, human iPSC-derived skeletal muscle, cell adhesion

## Abstract

In vitro culture longevity has long been a concern for disease modeling and drug testing when using contractable cells. The dynamic nature of certain cells, such as skeletal muscle, contributes to cell surface release, which limits the system’s ability to conduct long-term studies. This study hypothesized that regulating the extracellular matrix (ECM) dynamics should be able to prolong cell attachment on a culture surface. Human induced pluripotent stem cell (iPSC)-derived skeletal muscle (SKM) culture was utilized to test this hypothesis due to its forceful contractions in mature muscle culture, which can cause cell detachment. By specifically inhibiting matrix metalloproteinases (MMPs) that work to digest components of the ECM, it was shown that the SKM culture remained adhered for longer periods of time, up to 80 days. Functional testing of myofibers indicated that cells treated with the MMP inhibitors, tempol, and doxycycline, displayed a significantly reduced fatigue index, although the fidelity was not affected, while those treated with the MMP inducer, PMA, indicated a premature detachment and increased fatigue index. The MMP-modulating activity by the inhibitors and inducer was further validated by gel zymography analysis, where the MMP inhibitor showed minimally active MMPs, while the inducer-treated cells indicated high MMP activity. These data support the hypotheses that regulating the ECM dynamics can help maximize in vitro myotube longevity. This proof-of-principle strategy would benefit the modeling of diseases that require a long time to develop and the evaluation of chronic effects of potential therapeutics.

## 1. Introduction

Due to the increased interest in microphysiological systems (MPSs), in vitro test systems have gained traction in the scientific community for their use in understanding disease modeling, drug evaluation, and toxicity testing [1,2,3]. Human organ mimic platforms have the potential to recapitulate clinical responses and can serve to complement and possibly reduce the dependence on genetically engineered mice and other animal models. By integrating advances from induced pluripotent stem cell (iPSC) technology, in vitro organ mimics can now benefit from the potential to include the modeling of rare genetic disorders [4,5], developing personalized therapies for patients [6], and testing therapies [7]. Despite these advances, they still face multiple challenges [2,8]. One significant challenge is the maintenance of cell cultures over extended time periods, which is necessary for modeling diseases that require extended time to develop, for example, age-related diseases, or for testing the chronic effects of drug treatment [3,9]. One of the limitations for long-term culture is the premature surface release of cells from cultures. An example of a cellular model that can detach from surfaces prematurely is skeletal muscles due to their dynamic nature. Skeletal myofibers contract when functionally mature; however, these contractions will increase the risk of detachment and lead to premature termination of the culture. This is important as skeletal muscle models of longevity are needed due to their involvement in an increasing number of chronic or aging-related diseases, such as sarcopenia, diabetes, and Amyotrophic Lateral Sclerosis [10,11]. Therefore, this study chose to develop a strategy to extend the longevity of a human iPSC-derived skeletal muscle culture.

The adhesion of most mammalian cells to surfaces in vitro can be largely explained by cell–ECM interactions [12]. The ECM is a tissue-specific structure that in vivo is constantly undergoing remodeling and serves to enable cell attachment, structural and mechanical support, and cellular communication [13,14,15]. The initial stage of cell attachment to static surfaces involves light attraction via electrostatic interactions. Following this stage, the cells morphologically flatten via integrins, and finally, achieve complete focal adhesion through integrin–ECM bonding and reorganization of the actin cytoskeleton to allow the cells to adhere to the substrate [9]. However, a process that allows for cell mobility involves detachment, which is accomplished by degradation of the extracellular proteins regardless of the encapsulating system [16]. The metzincins family of matrix metalloproteinases (MMPs) have been identified as essential for this activity. These proteases depend on divalent metal ions for their catalytic activity and domain stability and are capable of degrading structural ECM proteins. MMPs influence diverse physiologic processes, from aspects of embryonic development to wound repair. When their activity is altered through insufficient spatiotemporal control, they influence and possibly inflict pathologic processes such as in inflammatory diseases and cancer metastasis [17].

Because of MMPs’ roles in the remodeling and breaking down of the ECM, they play a central role in the events of normal skeletal muscle development and regeneration [18,19]. While this study is mainly concerned with MMP-2 and MMP-9, MMP-3, 7, 10, and 14 are also expressed in in vitro skeletal muscle cells [20]. In the case of muscle development, studies have shown that different ratios of MMPs to TIMPs (tissue-specific inhibitors of metalloproteinases) are responsible for guiding the muscle cells through migration, fusion, and differentiation [21]. In vitro studies have shown that MMP-9 activity peaks in the first four days of culture prior to fusion, and almost entirely diminishes in the post-fusion and myotube period of the muscle cell progression. Conversely, MMP-2 activity remains constitutively active throughout all these stages. TIMP activity remains unnoticeable until post-fusion and peaks at the formation of myotubes, demonstrating that normal muscle fibers require strong inhibition against matrix remodeling to properly function [22,23]. In cases of muscle injury or inflammation, the satellite cells that are responsible for muscle regeneration are restricted to the basement membrane by ECM components. Upon degradation of the ECM, the detached and activated satellite cells migrate either underneath or through the basement membrane to reach the injury site. To repair the injured muscle, the cells differentiate into myocytes that fuse to form myotubes and later mature into myofibers [18]. The effect of MMP-9 has been localized to the inflammatory cells and its expression patterns are associated with satellite cell activation and myoblast proliferation [19,23,24]. MMP-2 levels hit a peak on Day 7 after injury and the effects are often associated with the concepts of fusion and differentiation due to its localization in the endomysium of regenerating fibers [23,24].

The physiological aging of cells in vitro relies heavily on environmental factors, such as those provided in culture media and in the extracellular matrix (ECM). The biomechanical properties of the ECM experience dynamic deterioration during aging, which will consequently affect cellular homeostasis. ECM dynamics is an emerging yet understudied hallmark of aging and longevity [25]. An in vivo study in *C. elegans* indicated that coordinated ECM remodeling through mechanotransduction is required and sufficient to promote longevity [26], suggesting targeting the ECM could be an effective avenue to increase the longevity of the muscle culture.

MMPs’ extensive involvement in skeletal muscle disorders during aging set them as attractive therapeutic targets. The observation of MMP inhibitory conditions noted decreasing accumulation of inflammatory cells, improved regeneration, contractile function, and myotube width [27,28,29,30]. MMP inhibition can be induced by using the following drugs: 4-hydroxy-Tempo (Tempol) (176141-5G, Sigma Aldrich, St. Louis, MO, USA) and Doxycycline (Sigma Aldrich, D9891-5G). Doxycycline is a tetracycline antibiotic that has been noted as a potential anti-cancer drug and is proving useful for MMP inhibition through chelating the zinc catalytic ion. Doxycycline-treated cells have exhibited higher adhesiveness, differentiation rates, and contractile force generation, as well as inhibited migration and matrix degradation [31,32,33,34,35]. Tempol is a superoxide scavenger that mimics superoxide dismutase activity and is used in the industry as an antioxidant. Oxidative stress is linked to muscle damage and inflammation and was experimentally shown to induce the expression of MMPs [36,37]. One of the mechanisms proposed for the observation involves NF-κB reduction [38]. NF-κB is upregulated under oxidative stress conditions and is linked to inducing MMP expression through binding to the *MMP* gene promoter. Tempol has been shown to reverse myogenesis inhibition caused by increased oxidative stress. Duansak et al. showed that Tempol treatment served to reduce NF-κB and lower MMP expression in their respective model [39]. As a positive control, Phorbol 12-myristate 13-acetate (PMA) (Fisher Scientific, AAJ63916MCR, Hampton, NH, USA) which is a phorbol ester known for its usage in cell cultures for tumor promotion was utilized. It has been linked to increased MMP expression on multiple occasions through a mechanism involving its binding to the Activator Protein-1 (AP-1), which is thought to induce the expression of *MMP* genes [40,41,42]. This study indicated that control of MMP activity can promote or decrease muscle cell longevity in culture. This has implications for the longevity of functional MPS devices and other in vitro models.

## 2. Materials and Methods

### 2.1. Surface Preparation

Glass substrates were roughened through a previously published acid-etching protocol [43]. Roughness was measured via atomic force microscopy (AFM). The substrates were then plasma-treated using a Harrick plasma cleaner (model PDC-32G) with high-purity oxygen gas for 20 min at 750 mTorr. A diethylene triamine (DETA) monolayer was formed, with a few modifications, on the glass substrate according to a previously described silane-coating protocol. DETA monolayer was verified via X-ray photoelectron spectroscopy (XPS) and glass coverslips were sterilized with 70% ethanol prior to any cell work.

### 2.2. Cell Culture

The iPSC line (ND41865) utilized in this study was purchased from the NIH Coriell Institute, and it was generated by reprogramming from the fibroblasts of a deidentified healthy subject. This study falls under NIH Exemption 4 as stated in that the research does not involve human subjects and 45 CFR part 46 does not apply. The obtained iPSCs were passaged over 10 times in mTesR medium (Stemcell Technology, Vancouver, BC, Canada) on matrigel-coated surfaces before being subjected to myoblast differentiation. Myoblast differentiation was performed strictly following a protocol published by Chal et al. [44]. The differentiation was carried out in our laboratory as described in Guo et al. [45]. Differentiated myoblasts from each batch of differentiation were cryopreserved in a liquid N_2_ tank and quality-controlled by standardized characterization assays before being utilized in experiments. For quality control, the myoblast purity was evaluated by quantifying the percentage of cells expressing representative myoblast markers MyoD1 and Pax7 after immunocytochemistry staining [45]. The myotube formation capability was evaluated by the measurement of the fusion index after fusion induction and testing of myofiber contractibility under electrical stimulation [10]. Only those batches showing a percentage of over 70% positive myoblast marker, a fusion index of 60% or higher, and demonstrating contractile myofibers were utilized in the experiment.

Myoblasts were plated on the previously described glass coverslips coated with Collagen-I in Myocult medium (Cat#5982, Stemcell Technology, Vancouver, BC, Canada). When the cells reached 60–70% confluency, they were switched to Dk-HI differentiation medium as described by Chal et al. [44]. Briefly, DK-HI medium consists of DMEM/F12 (Cat# 21041-025, ThermoFisher Scientific, Waltham, MA, USA), 15% vol/vol Knockout Serum Replacement (Cat# 10828-028, ThermoFisher Scientific), 10 ng/mL hepatocyte growth factor (HGF) (Cat# 315-23, Peprotech/ThermoFisher), 2 ng/mL insulin-like growth factor (IGF) (Cat# I1271, Sigma), 0.1 mM β-mercaptoethanol (Cat# 31350010, ThermoFisher Scientific), 1% MEM Non-Essential amino acids Solution (Cat# 11140050, ThermoFisher Scientific), and 100 nM dexamethasone (Cat# D4902, Sigma, Tokyo, Japan). After 48 h, the cultures were switched to the terminal maintenance medium Nbactiv4 (Cat# NB4-500, Brainbits, Springfield, IL, USA) and marked as “Day 0.” On Day 3, the cultures were dosed with the respective inducer or inhibitor of MMPs, and half medium change was performed every 3 days following that, with another drug dosing every 10 days following the functional testing of cells. Doxycycline and Tempol were administered as inhibitors in the dosages of 0.1/0.5/2.0 µg/ml and 50/100/125 µM., respectively. Phorbol 12-myristate 13-acetate was administered as an inducer at 5, 10, and 20 ng/mL concentrations.

### 2.3. Functional Testing

Muscle functionality was tested by delivering single pulses of 2 V at an increasing frequency (0.3, 0.5, 1.0, 2.0, and 4.0 Hz) directly to the skeletal muscle through a program written into LabVIEW software (2019 version). This test determined whether the skeletal muscle could contract in response to direct and controlled electrical stimulations. The muscle cultures were tested every 10 days starting on Day 10. In each culture, eight different regions of interest (ROIs) were selected from eight myotubes. Live video recordings of pixel differentiation were used to monitor the myotube contractions. Pixel differential is the subtraction of background noise from the pixel brightness changes caused by the contractions. This test quantifies the contraction fidelity, and the fatigue index of the myotube. Fidelity is calculated as the number of synchronized contractions at a given pulse divided by the number of stimulation pulses in a set duration. The fatigue index is calculated as $\left(1-\frac{area\;under\;curve}{peak\;force*time}\right)\;*\;100$. The protocol described reflects a modified version of the protocols described in Guo and Smith et al. [4].

### 2.4. Gelatin Zymography Assay

This assay was performed to measure gelatinase activity in the muscle culture by utilizing the gelatinase kit (Sigma Aldrich, MAK348-1KT). The protocol attached was followed with a few modifications. Medium from the respective wells were removed. Myotubes were collected using a cell scraper into microcentrifuge tubes, which were later spun down at 4100 rpm at 4 degrees for 5 min. The supernatant was aspirated, and the pellet was resuspended in the recommended volume of cell lysis buffer and the mixture was homogenized. The rest of the protocol steps were followed with each sample being read in triplicate, starting at t = 1 min to t = 30 min. With a total of 7 readings, each bar has an n = 21.

### 2.5. Statistical Analysis

All error bars on graphs represent the standard error of the mean (SEM). Statistical assessment was performed using one-way ANOVA in GraphPad Prism 9 and MS Excel (Version 1808). Graphs were created with GraphPad Prism 9 and MS Excel.

## 3. Results

### 3.1. Observation of Culture Dynamics Characterized by Phase Microscopy

Cultures were observed chronologically from Day 0 to Day 70 through phase microscopy (Figure 1). The myoblasts initially demonstrated elongated and spindle-like morphology. Myoblast fusion to form myotubes progressed from Day 0 to Day 10, typically reaching their optimal morphology around Day 10. As the muscle culture aged, the compact spindle morphology started to deteriorate into a web-like structure with more spaces in between, as observed on Day 30. By Day 40, abundant bundled-like structures were observed within the culture, indicating the retraction of myotubes off the surface. The number of myotubes by Day 50 were minimal and continued to decrease until Day 70, when virtually no myotubes were observed. Current culture conditions appear to maintain proper skeletal muscle morphology until approximately Day 30.

To test if MMP inhibition could preserve cell attachment for longer times, which could potentially serve as a viable way of increasing the longevity of iPSC skeletal muscle on the glass substrate, the effect of two MMP inhibitors, Tempol and Doxycycline, were tested in the muscle culture. The Doxycycline dosages tested were 0.1, 0.5, 1.0, and 2.0 µg/ml. The dose ranges were chosen based on the literature [31,32,46], and the optimal dose was determined based on its effect on muscle culture longevity in preliminary experiments, which indicated it to be 0.5 µg/ml (Appendix A). Similarly, the Tempol dosages tested were 25, 50, 100, and 125 µM, and the dosage of 100 µM was chosen as optimal for subsequent experiments (Appendix A). Figure 2A shows a comparison of the treated myotubes vs. untreated, indicating a morphological enhancement was maintained in the treated muscle cells in comparison to the untreated. At Day 35, the treated cultures indicated a more uniform distribution of myotubes across the surface, as opposed to the untreated culture, which showed gaps and some thinning of the myotubes. The spaces in the culture and bundled structures observed on Day 45 in the untreated culture were barely noticeable in the treated cultures. By Day 60, where the most significant difference was observed, the regular myotube morphology in the untreated culture had deteriorated, yet the morphology of the treated myotubes did not show a deterioration effect due to the culture age. Certain Doxycycline and Tempol treatments appeared to preserve proper myotube morphology until approximately Day 65, while the untreated cultures indicated deterioration as early as Day 35. Quantification of the cultures preserved over time indicated that 40% of the untreated cultures were peeled by Day 50 with repetitive functional testing (tested every 10 days by electrical stimulation), while the Doxycycline treatment indicated about 20% peeling, and the Tempol treatment had almost 0% peeling at Day 50, indicating at least 20 days prolongation of the muscle culture compared to the regular condition, with enhanced functional characteristics.

### 3.2. Effects of Doxycycline and Tempol Characterized by Skeletal Muscle Functional Testing

To ensure the functionality of the treated myotubes was preserved along with the morphology in the cultures treated with MMP inhibitors, the contractile function of the myotubes in response to field electrode stimulation at varied frequencies was tested. All myotubes were subjected to stimulated contraction at frequencies of 0.33, 0.5, 1.0, 2.0, and 4.0 Hz on Day 10, 20, 30, 40, and 50 post-differentiation, and compared to the untreated myotubes. Figure 3 shows the representative raw stimulation–contraction traces at Day 30, the red lines indicate the stimulation at increasing frequencies, and the blue lines indicate the muscle contraction response to the stimulation at given frequencies. While the amplitude of contraction of the treated myotubes at 0.5 μg/mL Doxycycline and 100 μM Tempol was close to 8000, the amplitude of the untreated myotube contraction was slightly below 4000, which was approximately half of the treated myotube amplitudes. The amplitude is proportional to the contractile force generation of the myotubes. Figure 3A at 4 Hz shows minimal contraction, initially, with background noise and some spontaneous contractions towards the end, where the amplitude continues to rise without any direct stimulation. This spontaneous contraction is one of the observations most representative of the untreated condition at Day 30, with the other being unresponsiveness to stimulation. Quantification of these functional data was performed to determine the differences between the treated and untreated myotubes and their temporal manifestation. Two functional parameters were analyzed: the fatigue index, which measures the extent of myofiber fatigue during a period of high-frequency stimulation, and fidelity, which measures the percentage of stimulation-induced myofiber contractions out of the total number of stimulations. Both were evaluated at multiple stimulation frequencies and the detailed explanation of these two parameters is described in [4]. As shown in Figure 4A, there is a significant reduction in the fatigue index in the treated myotubes compared to the untreated myotubes starting at Day 10 through 50. The difference is more prominent in later-stage cultures at D40 and D50, suggesting better preservation of muscle function by MMP inhibition. This analysis also reaffirmed the optimal dosage for Tempol (100 μM) and Doxycycline (0.5 μg/mL), which demonstrates the lowest fatigue index compared with other dosages. Figure 4B compares the contraction fidelity of the treated vs. untreated. The result displays a non-statistically significant change in the fidelity across myotubes, and that trend appears to remain constant over 50 days. The functional data support the hypothesis that while preserving morphology, Doxycycline and Tempol treatments can reduce muscle fatigue and maintain their contraction fidelity for longer periods of time.

### 3.3. Effect of PMA on the Longevity of Muscle Culture Characterized by Phase Microscopy

To confirm that the observations regarding morphology and functionality were preserved due to the mechanism of inhibiting MMP activity, cells were treated with an MMP inducer, PMA, to check for the reversal of these observations. The PMA was dissolved in DMSO prior to dosing, so a vehicle control of DMSO-treated culture was included. The final concentration of DMSO (*v*/*v*) in the well did not exceed 0.1%, with the 10 ng/mL treatment having 0.08% DMSO (*v*/*v*) and the 15 ng/mL having 0.1% DMSO (*v*/*v*). Figure 5A shows the cultures treated with 10 ng/mL of PMA. Cultures on Day 10 indicated a healthy morphology with minor spacing. The myotubes started to show gradual thinning and dramatic spacing with more bundle structures and aggregation between Days 14 and 26. On Days 30 and 34, small myotube fragments were observed, confirming the peeling of some myotubes, with Day 38 indicating a noticeable decrease in muscle mass, while myotubes were better preserved in the DMSO control condition. The effect of the PMA on the preservation of the muscle culture under two concentrations was quantified and is illustrated in Figure 5B. By Day 20, a 33.4% decrease in the number of myotubes was observed in the higher PMA concentration with the remaining 66.6% of cultures peeling by Day 30, with the control culture still having 100% of the myotubes remaining. Cultures treated with 10 ng/mL PMA dropped by 33.4% by Day 30 and showed a significantly larger fatigue index than the control, with the remaining 66.6% of cultures peeling by Day 40, while the control culture still maintained about 66% of the initial number of myotubes.

### 3.4. Effects of PMA Characterized by Skeletal Muscle Functional Testing

Functional testing of the PMA-treated cultures was carried out as above to evaluate the effect of PMA in terms of the fatigue index (Figure 6A) and contraction fidelity (Figure 6B) by comparing with the vehicle control. Testing of the PMA-treated cultures lasted 20 days for the 15 ng/mL PMA concentration, and 30 days for the 10 ng/mL PMA concentration, mainly due to a dose-dependent increase in peeling. In terms of the fatigue index in Figure 6A, Day 10 showed no significant differences between the untreated cultures and different PMA-treated cultures. Day 20 indicated a slight increase in the fatigue index proportional to the PMA concentrations, although it was not significant. By Day 30, a significant increase in the fatigue index was observed compared to the untreated cultures, with the result for 15 ng/mL PMA not shown since all the cultures peeled under this treatment condition. In terms of the fidelity in Figure 6B, no significant change in the fidelity was observed by the time the 15 ng/mL PMA-treated cultures had peeled. Day 40 cultures were not analyzed since all PMA cultures had peeled by then.

### 3.5. Gelatinase Activity Assay

To verify the hypothesis that the preserved morphology and functionality in the Tempol- and Doxycycline-treated cultures and a degradation effect in the PMA cultures were at least in part due to the MMP alterations, the MMP enzymatic activity under these conditions was evaluated by the gelatinase assay. Gelatinases (MMP-2 and MMP-9) are from the MMP family and are associated with the degradation of collagen, an essential component of the ECM [47]. Cultures were treated with the MMP inhibitors Doxycycline and Tempo as well as the MMP inducer, PMA. Figure 7 indicates a comparison of MMP activity across the treated cultures. The MMP activity of the untreated culture is almost equal to the vehicle control (culture treated with DMSO), while the culture treated with PMA had significantly increased MMP activity compared to the untreated. Both inhibitors, Doxycycline and Tempol, significantly reduced the MMP activity in their respective cultures compared to the untreated culture. The cultures treated with both an inhibitor and an inducer were to investigate a rescue/blocking mechanism. This contrast of gelatinase activity between the MMP inhibitor and activator conditions correlates well with the phase image observations in Figure 2A and Figure 5A. The most notable difference was between cultures treated with PMA and the inhibitors, and while the PMA treatment indicates peeling and thinning of myotubes, the inhibitor-treated cultures show the expected uniform myofiber-like morphology with the highest density of myotubes.

## 4. Discussion

The premature peeling of myotubes from in vitro surfaces limits the application of in vitro muscle systems for the modeling of chronic diseases, especially those associated with aging. The peeling issue is further exacerbated with repeated functional testing. The goal of this study was to leverage the regulation of ECM dynamics, specifically through the action of MMPs, as a tool to maximize skeletal muscle longevity in vitro. To this end, a skeletal muscle model of increased longevity was presented with functional readouts until Day 50. Two drugs with different mechanisms of MMP inhibition were investigated for their effects on skeletal muscle adhesion. Doxycycline is an activity inhibitor that works to chelate the zinc atom responsible for the catalytic activity of the proteinases [48]. Tempol is an antioxidant that serves to reduce the oxidative stress that increases naturally as cells age. While the presented functional readouts went until Day 50, the cultures were able to be maintained up to Day 68 and their population morphology was monitored over time using phase microscopy. The phase images indicate a preservation of myotubes for up to 60 days in the inhibitor-treated cultures, compared to the untreated cultures, which displayed peeling as early as Day 35, while the PMA-treated cultures displayed peeling as early as Day 10. Myoblasts deposited on the surface morphologically flatten and spread out over time by means of integrin binding and extracellular matrix interactions until myotubes fuse. Typically, an optimal mature myotube morphology is reached at Day 10, indicating why the functional testing started on Day 10. Untreated myotubes begin exhibiting bright phase images elucidating the 3D contour and as they start to age, spacing between myotubes is observed, accompanied by bundle structures and retraction from the surface. Physiologically, this loss of muscle mass is associated with muscle aging, as reflected in sarcopenia conditions [49,50]. The fast progression observed from Day 30 to Day 60 Days in Figure 1 can also be explained by the increased expression of apoptotic and inflammatory factors as the muscles age [51]. MMP-9 has been linked to inflammatory factors in multiple publications [23,24,27], and its upregulation would explain the fast ECM turnover and deterioration of the muscle with time in the culture. The gel zymography data further supported the hypothesis that the observed difference in the cultures was linked with the differential activity of MMP found in the respective cultures. As indicated, there was a statistically significant increase in MMP activity in the PMA-treated culture, and statistically significant decrease in MMP activity in the PMA inhibitor-treated cultures. If MMPs are upregulated, ECM degradation is expected, and in in vitro cultures, that typically represents the degradation of the attractants keeping the cells adhered to the surface. The inhibitor-treated cultures both showed a significantly reduced MMP activity compared to the untreated cultures, in agreement with the previous literature [27,31,32,35]. In line with the reduced MMP activity, the inhibitor-treated cultures at Day 42 demonstrated a robust, uniform culture with the appearance of healthy myotubes. This confirms the hypothesis that inhibition of ECM degradation by MMPs can enable a retention of the ECM attachments of the muscle culture for longer periods. This is also consistent with the previous literature stating that the use of inhibitors assists in cell adhesion, decreased migration, and matrix degeneration [27,31,32]. In the cultures treated with both inhibitors and inducers, an MMP activity similar to the untreated culture is observed. It appears that the inducer and inhibitor cancel each other’s activity. While Tempol was initially expected to neutralize PMA effects, since both chemicals play a role in the transcriptional control of MMPs, Doxycycline seemed to mask the effects of PMA slightly better, despite being an activity inhibitor.

Functional testing of myotubes was performed to ensure that the longevity of myotubes achieved was accompanied by a retention of function. Myotube functional assessment is essential for disease modeling and drug toxicology evaluation. Generally, there is a tradeoff between the functional evaluation of myotubes and their longevity. The more frequently stimulated the myotubes are, the higher the chance of their peeling. Certain treated cultures consistently displayed a lower fatigue index and higher contraction fidelity up to Day 50 with repeated testing. While the data from Days 10, 20, and 30 are based on at least 10 replicates, about 40% of the untreated cultures had peeled by Day 40 and 55% by Day 50 of testing, which is reflected in a decrease in the number of ns for those days. Treated myotubes at certain dosages indicated a significantly reduced fatigue index. In terms of contraction fidelity, while the untreated myotubes more frequently displayed asynchronous contractions (Figure 3), which included spontaneous contractions or a lack of response under direct stimulation, the treated myotubes showed a non-significant trend for a higher degree of consistent contraction with increasing frequency. The Doxycycline-treated cultures (2.0 µg/mL) consistently showed lower fidelity than the untreated myotubes. That may be attributed to the toxicity effects on the culture near that dosage. The traces in Figure 3 show the treated cultures contracting at almost double the amplitude generated by the untreated cultures. This correlates to a higher force generated by the culture, as seen in the previous literature [52]. Conversely, the functional data supported the hypothesis that early detachment and distorted morphology was caused by the PMA treatment of the myotubes. In general, a reduced contractile force generation, higher fatigue index, and decrease in muscle mass are all hallmarks of aging, which were displayed in our model, and they were counteracted by the PMA inhibitors, and accelerated by the inducer [50,53].

This study indicates that an ECM-targeting strategy increased the cell attachment of human-derived skeletal muscle cells in culture by comparison to those under standard culture conditions, and enabled muscle cell culture lasting up to 80 days. This improvement will significantly increase the physiological/pathological domain that can be modeled in the in vitro systems, which enables better recapitulation of aging or aging-related diseases in vitro and allows for the testing of chronic drug effects. A similar strategy can be applied to other cell culture systems to promote longevity, especially for contractable cells, with the understanding that the methodology details may need to be adjusted for each cell culture at different stages due to the specific variation in ECM targets in different contexts. Additionally, although a large body of evidence indicating that the ECM can be a hallmark of aging is emerging, and the data in invertebrates are promising, direct experimental proof that activating ECM homeostasis is sufficient to slow aging in mammals is lacking, as reviewed in [25]. This study provides experimental evidence for increasing the longevity of human skeletal muscle in cultures by modifying ECM dynamics.

## 5. Conclusions

This study investigated MMP inhibition as a strategy to maximize myotube longevity in vitro for microphysiological systems, which generated a functional human skeletal muscle model of extended longevity. Myotubes were maintained on the surfaces for up to 80 days without repeated functional testing, and 50 days with repeated testing, which compares with ~18 days following our previous protocols. The treated myotubes indicated a statistically significant reduction of fatigue index over the course of repeat testing. The MMP zymography assay confirmed that the phenotypic changes observed in the muscle culture were accompanied by corresponding changes in the MMP activity. These muscle cultures of increased longevity can be adapted into complex microphysiological systems to obtain more robust cultures needed for assessments of long-term therapeutics and aging disease models. This work also provides experimental proof for the principle of promoting cell longevity through modifying ECM dynamics in mammalian cells. Both the developed skeletal muscle culture of extended longevity and the concept proof for targeting the ECM to promote cell adhesion will contribute significantly to progress in human-derived modeling. This field is important to bridge the translation gap in disease modeling and therapeutic development and will help shorten the time and cost often associated with drug discovery and long clinical studies.

## Figures and Tables

**Figure 1 biomedicines-12-00856-f001:**
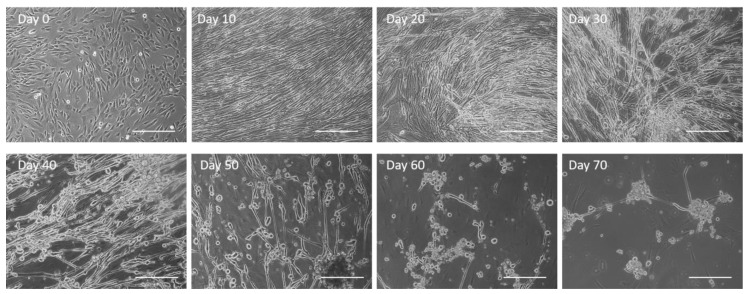
Untreated human skeletal muscle progression over a period of 70 days with deterioration in morphology observed as early as Day 30 and complete cell peeling by Day 60. Scale bar: 100 µm.

**Figure 2 biomedicines-12-00856-f002:**
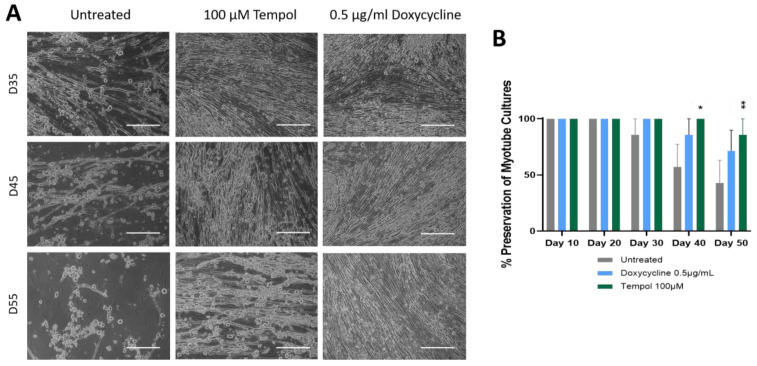
A comparison of treated myotubes vs. untreated myotubes over a period of 55 days. (**A**) Phase images. Treated cells show retention of morphology and muscle density until Day 55 while untreated cultures indicate signs of deterioration and peeling as early as Day 35. Scale bar: 100 µm. (**B**) Analysis of the preservation of myotube cultures by quantifying the percentage of completely peeled coverslips to the original number of coverslips from each condition in a batch. The statistical comparison was performed between each treated condition to the untreated culture. N = 7 batches of experiments, one-way ANOVA, mean ± SEM. * *p* < 0.05, ** *p* < 0.01.

**Figure 3 biomedicines-12-00856-f003:**
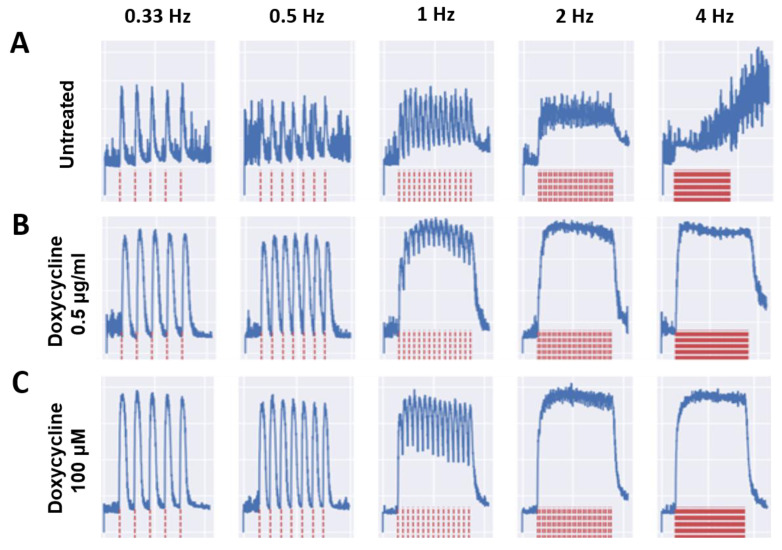
Sample stimulation–contraction traces from Day 30 under different treatment conditions. The electrical stimulation was applied at different frequencies ranging from 0.33 Hz to 4 Hz, as illustrated in red traces at the bottom. The myofiber contraction traces upon each stimulation are shown in blue at the top. (**A**) Untreated culture, (**B**) Doxycycline 0.5µg/mL culture, and (**C**) Tempol 100 µM culture.

**Figure 4 biomedicines-12-00856-f004:**
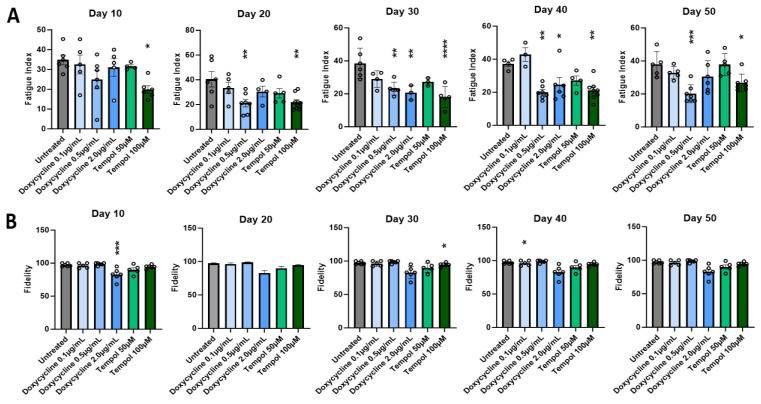
Quantitative analysis of the effect of MMP inhibition on myofiber contractile function for cultures every 10 days across 50 days. (**A**) Fatigue index. (**B**) Fidelity. Data are from 6 batches of experiments. The statistical comparison was performed between each treated condition with the untreated culture. One-way ANOVA, mean ± SEM. * *p* < 0.05, ** *p* < 0.01, *** *p* < 0.001, **** *p* < 0.0001.

**Figure 5 biomedicines-12-00856-f005:**
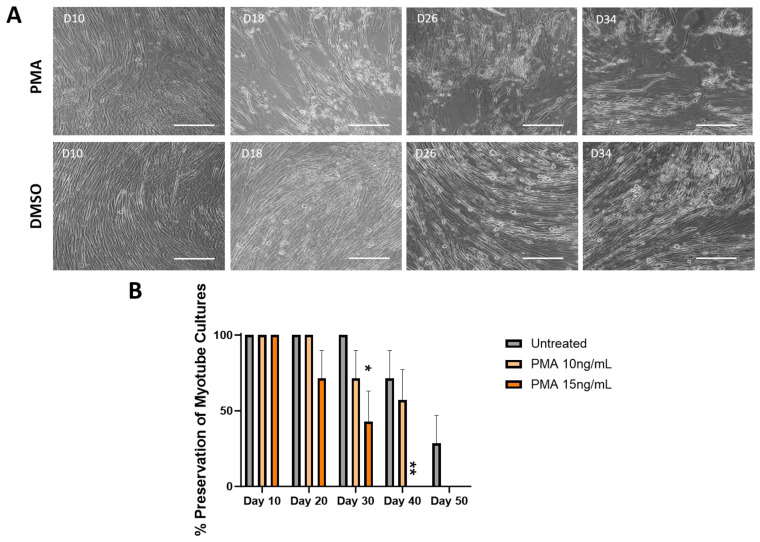
Effect of MMP activation by PMA on the percentage of myotube cultures remained intact over 50 days. (**A**) Sample phase microscopy images for cultures treated with 10 ng/mL PMA compared to the DMSO vehicle control demonstrating the expediated cell peeling under the PMA condition, while DMSO-treated cultures indicated retention of morphology through day 34, closely resembled the progression of untreated cultures. Scale bar: 100 µm. (**B**) Quantification of the cell preservation under different treatment conditions by quantifying the percentage of completely peeled coverslips to the original number of coverslips from each condition in a batch. Higher dosages of PMA displayed 66.6% myotube peeling by Day 30 and 100% myotube peeling by Day 40, while 10 ng/mL treatment indicated about 33.4% myotube peeling by Day 30 and 66.4% by Day 40. Vehicle control cultures showed the longest preservation of cultures over time. The statistical comparison was performed between each treated condition with the vehicle control. N = 7 batches of experiments. One-way ANOVA. Mean ± SEM. * *p* < 0.05, ** *p* < 0.01.

**Figure 6 biomedicines-12-00856-f006:**
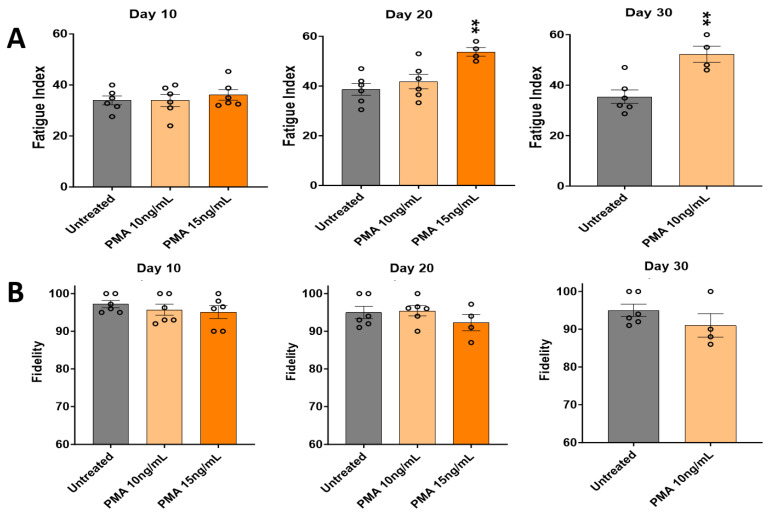
Effect of PMA on the contractile function of iPSC-derived SKM across 30 days. (**A**) Fatigue index. (**B**) Fidelity. Results are from 6 batches of experiments. Each circle represents one replicate. There were fewer than 6 replicates in some of the later groups, indicating the loss of some systems under PMA conditions. The statistical comparison was performed between each treated condition with the untreated condition. One-way ANOVA. Mean ± SEM. ** *p* < 0.01.

**Figure 7 biomedicines-12-00856-f007:**
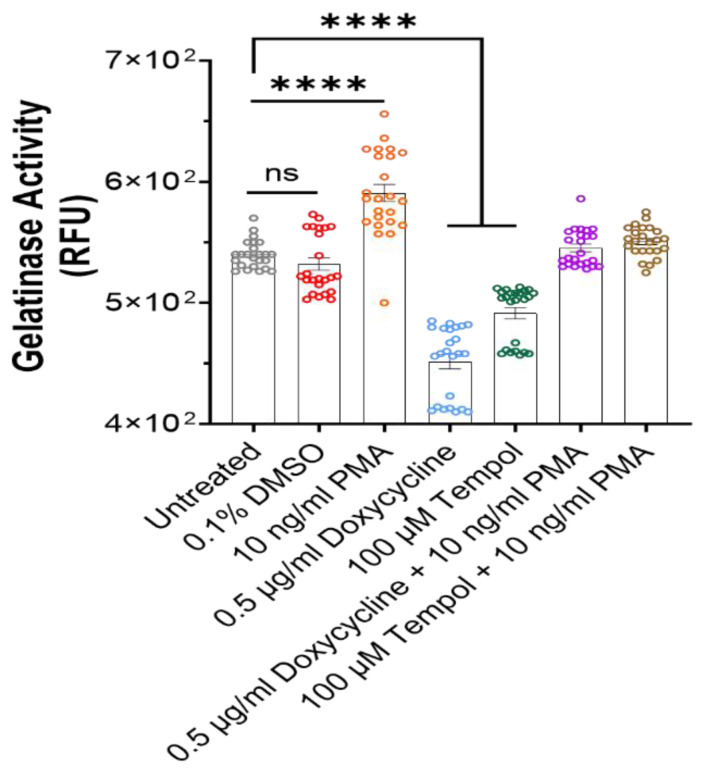
Visualization of gelatinase activity in the medium across all the cultures at Day 42 utilizing gelatinase activity assay. PMA-treated cultures showed a significantly higher activity compared to untreated cultures, while inhibitor-treated cultures showed significantly reduced activity. Combination of the gelatinase inhibitor and activator canceled out the effect. Data represented from 24 replicates collected from 3 batches of experiments. One-way ANOVA. Mean ± SEM. ns, no significant difference, **** *p* < 0.0001.

## Data Availability

All the data for this work will be available upon proper request from the author.

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
