# Peer review of "Inhibition of Metalloproteinases Extends Longevity and Function of In Vitro Human iPSC-Derived Skeletal Muscle"

_biomedicines, 2024, doi:10.3390/biomedicines12040856_

Round 1

Reviewer 1 Report

Comments and Suggestions for Authors

TITLE

            I would like to suggest the title was ‘Metalloproteinases inhibition extends in vitro longevity and functionality of hiPSC- derived skeletal muscle cells to suit better.

ABSTRACT

            The abstract is well written; just an English review is required. I mentioned blow that the skeletal muscle use in this study is vague. The experimental design is clear, but among all iPSC-derived cells, why choosing skeletal muscle cells? I think the state of significance for the use of these cells is required, otherwise seemed that you randomly used these cells.

INTRODUCTION

            The Introduction is well structured, detailed and brings the required background to understand the importance of MMPs to the in vitro systems related to iPSC culture. My only critics is that why human iPSC- skeletal muscle were chosen to develop the study. Among all iPSCs-derived cell types, why the group chose skeletal muscle to study MMPs influence on cell longevity. The background is a little vague; once you start talking about microphysiological systems in general and then skeletal muscle disorders suddenly appear. Check that.

MATERIALS AND METHODS

- Ethical approval for the study is missing, as well as the information related to the source of human cells used for cell reprogramming and iPSC establishment;

- Myoblasts differentiation, even following an established protocol must be validated and characterized to carry out the studies, so I request the characterization assays with their respective controls, even as Supplementary Material, but they are mandatory for this sort of publication.

RESULTS

- I believe there was a typing error in the Results section, once the content from line 314 to 325 should come before Figure 1. Just correct that please;

- Which was the quantification method used to assess the comparison between treated myotubes vs. untreated myotubes in the light microscopy? How many replicates did you performed? It si not clear.

- Please, improve the quality of the images, provide the characterization data of the cells and the zymography gels as well.

DISCUSSION

            English in this section must be revised. The Discussion is elaborated and correlates all the findings with the literature.

CONCLUSIONS

            English in this section must be revised. The conclusion is a little long, but synthetizes all the findings of the study.

Comments on the Quality of English Language

Minor English revision. 

Author Response

Reviewer 1

TITLE I would like to suggest the title was ‘Metalloproteinases inhibition extends in vitro longevity and functionality of hiPSC- derived skeletal muscle cells’ to suit better.

Response: Sorry, but we think the original title describes the research better.

ABSTRACT The abstract is well written; just an English review is required. I mentioned blow that the skeletal muscle use in this study is vague. The experimental design is clear, but among all iPSC derived cells, why choosing skeletal muscle cells? I think the state of significance for the use of these cells is required, otherwise seemed that you randomly used these cells.

Response: The abstract was revised. The rational for choosing skeletal muscle was added in both the abstract and the introduction, but in brief, they are contractile cells which would create the most stress on the ECM.

INTRODUCTION - The Introduction is well structured, detailed and brings the required background to understand the importance of MMPs to the in vitro systems related to iPSC culture. My only critics is that why human iPSC- skeletal muscle were chosen to develop the study. Among all iPSCs-derived cell types, why the group chose skeletal muscle to study MMPs influence on cell longevity. The background is a little vague; once you start talking about microphysiological systems in general and then skeletal muscle disorders suddenly appear. Check that.

Response: Revised accordingly, but as above, they are contractile.

MATERIALS AND METHODS - Ethical approval for the study is missing, as well as the information related to the source of human cells used for cell reprogramming and iPSC establishment; - Myoblasts differentiation, even following an established protocol must be validated and characterized to carry out the studies, so I request the characterization assays with their respective controls, even as Supplementary Material, but they are mandatory for this sort of publication.

Response: Correspondent information has been added.

RESULTS - I believe there was a typing error in the Results section, once the content from line 314 to 325 should come before Figure 1. Just correct that please;

Response: Corrected. Thank the reviewer for pointing this out, it has been corrected.

Which was the quantification method used to assess the comparison between treated myotubes vs. untreated myotubes in the light microscopy? How many replicates did you performed? It si not clear. - Please, improve the quality of the images, provide the characterization data of the cells and the zymography gels as well. –

Response: As stated in the figure legends, the quantification accompanying the phase images is conducted “by quantifying the percentage of completely peeled coverslips to the original number of coverslips from each condition in a batch”. The statistic information has been added to correspondent figure legends. The image quality has been improved and the original figures have been replaced. For the gelatinase anyalysis, a fluorometric gelatinase kit was used, and no zymography gel was involved.

DISCUSSION - English in this section must be revised. The Discussion is elaborated and correlates all the findings with the literature.

Response: Reviewed.

CONCLUSIONS English in this section must be revised. The conclusion is a little long, but synthetizes all the findings of the study.

Response: The conclusion has been modified to address the reviewers comments.

Reviewer 2 Report

Comments and Suggestions for Authors

This article discusses that by regulating the dynamics of the extracellular matrix (ECM), in particular by inhibiting matrix metalloproteinases (MMPs), it is possible to prolong the attachment of cells, especially skeletal muscle cells, in in vitro cultures. The study demonstrates that inhibiting MMPs leads to longer periods of cell adherence to culture surfaces, resulting in reduced fatigue index and maintained fidelity compared to untreated cultures. This approach allows for the extension of muscle cell culture longevity, enabling studies lasting up to 80 days. The implications of this research include improved disease modeling and drug testing capabilities, particularly for conditions requiring long-term culture observation or the evaluation of chronic effects or toxicity of potential therapeutics.

The article can be accepted for publication after the following comments have been eliminated:

1. It is necessary to improve the quality of the Figure 3.

2. Correct Figure 5B. It's stretched out.

Author Response

Reviewer 2

The article can be accepted for publication after the following comments have been eliminated:

  1. It is necessary to improve the quality of the Figure 3.

Response: The figure has been modified to address the reviewers comments.

  1. Correct Figure 5B. It's stretched out.

Response: The figure has been modified to address the reviewers comments.

Reviewer 3 Report

Comments and Suggestions for Authors

1. Broaden the introduction with a brief overview of in vitro culture challenges and the role of ECM dynamics.

2. Provide stronger justification for the hypothesis that inhibiting MMPs extends cell attachment, including references to previous studies.

3. Offer more methodological details, particularly on MMP inhibitors/inducers' selection and usage.

4. Improve data presentation with additional quantitative data, statistical analysis, and clear visualizations to underline key findings.

5. Discuss the study's limitations more thoroughly, including potential biases and the generalizability of the MMP inhibition strategy.

6. Further elaborate on the potential applications of your findings, particularly in disease modeling and drug testing scenarios, and discuss scalability.

7. Update and broaden references to include recent studies on ECM dynamics and cell culture longevity.(https://doi.org/10.1016/j.compbiomed.2024.108099)

8. Conduct a comprehensive technical and language review for clarity, readability, and consistency in terminology. 

Comments on the Quality of English Language

Minor editing of the English language required

Author Response

Reviewer 3

  1. Broaden the introduction with a brief overview of in vitro culture challenges and the role of ECM dynamics.

Response: This has been revised as requested.

  1. Provide stronger justification for the hypothesis that inhibiting MMPs extends cell attachment, including references to previous studies.

Response: We have tried to gather relevant literature as included in the introduction. However, no previous studies can be found that directly indicating inhibiting MMP can extend cell attachment. Actually, this is a novelty from this study.

  1. Offer more methodological details, particularly on MMP inhibitors/inducers' selection and usage.

Response: The methodology details have been provided in the introduction, results or supplementary data.

  1. Improve data presentation with additional quantitative data, statistical analysis, and clear visualizations to underline key findings.

Response: Statistical information has been added to correspondent figure legends.

  1. Discuss the study's limitations more thoroughly, including potential biases and the generalizability of the MMP inhibition strategy.

Response: This has been added to the discussion.

  1. Further elaborate on the potential applications of your findings, particularly in disease modeling and drug testing scenarios, and discuss scalability.

Response: This has been added to the discussion.

  1. Update and broaden references to include recent studies on ECM dynamics and cell culture longevity. (https://doi.org/10.1016/j.compbiomed.2024.108099)

Response: The recommended reference is not relevant. But more recent relevant references are included in introduction as suggested.

  1. Conduct a comprehensive technical and language review for clarity, readability, and consistency in terminology.

Response: This has been reviewed and updated.